# Efficacy of Autologous Intrauterine Infusion of Platelet-Rich Plasma in Patients with Unexplained Repeated Implantation Failures in Embryo Transfer: A Systematic Review and Meta-Analysis

**DOI:** 10.3390/jcm11226753

**Published:** 2022-11-15

**Authors:** Muzi Li, Yan Kang, Qianfei Wang, Lei Yan

**Affiliations:** 1Center for Reproductive Medicine, Shandong University, Jinan 250100, China; 2Key Laboratory of Reproductive Endocrinology of Ministry of Education, Shandong University, Jinan 250100, China; 3Shandong Key Laboratory of Reproductive Medicine, Jinan 250012, China; 4Medical Integration and Practice Center, Shandong University, Jinan 250100, China; 5Gynecology Department, Reproductive Hospital Affiliated to Shandong University, Jinan 250001, China; 6Obstetrics Department, Shandong Provincial Maternal and Child Health Care Hospital, Jinan 250001, China

**Keywords:** platelet-rich plasma, repeated implantation failures, embryo transfer, clinical pregnancy rates, intracytoplasmic injection, in vitro fertilization

## Abstract

(1) Background: Controversial conclusions have been made in previous studies regarding the influence of autologous platelet-rich plasma (PRP) in the reproductive outcomes of women with repeated implantation failures (RIF) who are undergoing embryo transfer (ET). (2) Methods: This study aimed to evaluate the effect of PRP intrauterine infusion in patients with unexplained RIF, who are undergoing in vitro fertilization (IVF) or intracytoplasmic injection (ICSI), by a systematic review and meta-analysis. (3) Results: A fixed-effects model was used, and 795 cases and 834 controls were included in these studies. The pooling of the results showed the beneficial effect of PRP which were compared with those of the control in terms of the clinical pregnancy rates (n = 10, risk ratio (RR) = 1.79, 95% confidence intervals (CI): 1.55, 2.06; *p* < 0.01, I^2^ = 40%), live birth rates (n = 4, RR = 2.92, 95% CI: 2.22, 3.85; *p* < 0.01, I^2^ = 83%), implantation rates (n = 3, RR = 1.74, 95% CI: 1.34, 2.26; *p* < 0.01, I^2^ = 0%), and positive serum β-HCG 14 days after the ET (n = 8, RR = 1.77, 95% CI: 1.54, 2.03; *p* < 0.01, I^2^ = 36%). However, we did not find that the miscarriage rates indicated a significant difference between the two groups (n = 6, RR = 1.04, 95% CI: 0.72, 1.51; *p* = 0.83, I^2^ = 0%). (4) Conclusions: The findings of this systemic review and meta-analysis suggest that PRP appears to improve the results of IVF/ICSI treatments in the cases of unexplained RIF.

## 1. Introduction

A considerable proportion of couples worldwide suffer from infertility [1]. The common reasons for female infertility issues include ovulation disorders, fallopian-related disorders, uterine disorders, and unexplained infertility [2]. Although assisted reproductive technology (ART) has rapidly developed in recent years, the causes and treatments of repeated implantation failures (RIF, recurrent implantation failures) continue to plague reproductive specialists. There is no accepted formal definition for RIF due to the fact that RIF was initially considered to be a rather heterogeneous entity. Some studies have defined it as a failure of the implant after three or more embryo transfers (ETs) with high-quality embryos [3,4]. It was, however, also accepted that RIF was considered to be a disorder of infertility women who had undergone at least two ET failures [5,6]. Even so, quite a few specialists have suggested a more complete working definition, taking into account the maternal age, the number of embryos that were transferred, and the number of cycles that were completed [7,8]. The accumulative data have clarified that most of the etiology does not have the evidence base for a generalized application to be suggested by the relevant societies. The etiology of RIF was currently attributed to the dysfunction of the embryo and the endometrium, and Antonis Makrigiannakis et al., also divided it into several factors, namely anatomy, immunology, dysbiotic microbiota, and unexplained reasons, etc., in their review [2]. Moreover, this team has also described some of the main treatment protocols including endometrium injury, human chorionic gonadotropin, peripheral blood mononuclear cells, and platelet-rich plasma (PRP). RIF is a constant challenge in ART with it being a burden on health providers and infertile couples.

The extraction protocols of platelet rich-plasma also remain inconsistent, with there being no consensus around the world. It is generally defined as an autologous blood-derived concentrate of the platelets from the peripheral blood that has a platelet count that is 35 times higher than the baseline concentration with growth factors and other cytokines such as transforming growth factor beta (TGF-β) and interleukin-1β (IL-1β) [9,10,11,12]. Presently, PRP is widely used in knee osteoarthritis, erectile dysfunction, medical dermatology, periodontal regeneration, and facial rejuvenation [13,14,15,16].

In reproductive medicine, a poor ovarian reserve, premature ovarian failure, and a thin endometrium have been the main areas of the research of PRP by intraovarian injection or intrauterin infusion [17,18,19]. Studies found that PRP had high growth factor and cytokine concentrations, which are considered to be very important in cell proliferation, chemotaxis, cell differentiation, regeneration, and angiogenesis [20,21]. Fady I Sharara et al., reviewed the previous literature on the effects of autologous PRP in reproductive medicine, finding that PRP can increase the endometrial thickness in thin endometrium [22]. These reasons may explain why PRP could improve the implantation outcomes and be beneficial for embryo transfer.

There has been a surge in high-level studies investigating PRP for implantation failures. Therefore, in this systematic review and meta-analysis, we aimed at investigating the effect of the intrauterine infusion of autologous PRP in women with unexplained RIF who are undergoing IVF/ICSI cycles.

## 2. Methods

By complying with the guidelines of recommendations of the Cochrane Handbook for Systematic Reviews of Interventions and Preferred Reporting Items for Systematic Reviews and Meta-Analyses (PRISMA) to search for the relevant studies that had been published in the medical literature up to the time of this research, this systematic review and meta-analysis gathered statistics and evaluated the efficacy of autologous PRP intrauterine infusion on the pregnancy results for patients with RIF of unknown causes which were compared those who underwent no treatments or other treatments. Clinical pregnancy is the first outcome, and it is defined as the presence of a fetal heartbeat or the gestational sac in transvaginal ultrasonography 4–6 weeks after the embryo transfer. Live birth is defined as the delivery of a live-born child after 24 weeks of gestational age. Miscarriage is defined as s a fetal loss before 20 weeks of gestation.

### 2.1. Literature Search

We identified potential studies by searching Medline (PubMed), Embase, Cochrane Library, and Web of Science (WOS). Additionally, we identified eligible literature searches by using the references of published articles. The search terms included: (“RIF” OR “Repeated implantation failure” OR “Implantation failure” OR “Recurrent implantation failure” AND “Platelet-rich plasma” OR “PRP” OR “Autologous platelet-rich plasma” OR “Platelet-rich plasma gel” OR “Plasma, Platelet-Rich” OR “Platelet Rich Plasma”). Among these terms, platelet-rich plasma is a MESH, and the others are free terms. The terms were searched in the title and abstract parts of the studies. The database-specific indexing terminology is listed as ((((((Plasma, Platelet-Rich[Title/Abstract]) OR (Platelet Rich Plasma[Title/Abstract])) OR (PRP[Title/Abstract])) OR (Autologous Platelet-Rich Plasma[Title/Abstract])) OR (Platelet-Rich Plasma Gel[Title/Abstract])) OR (“Platelet-Rich Plasm”[Mesh])) AND ((((Repeated implantation failure[Title/Abstract]) OR (implantation failure[Title/Abstract])) OR (recurrent implantation failure[Title/Abstract])) OR (RIF[Title/Abstract])).

### 2.2. Selection (Inclusion and Exclusion) Criteria

Two independent reviewers censored the titles and abstracts of the identified studies. Thereafter, the selected studies were thoroughly and completely read in order to made a decision about their inclusion or exclusion from this meta-analysis. Given that there is currently no consensus on the definition of RIF, we included medical records with a clear diagnosis in the studies. The randomized controlled trials and cohorts that underwent the PRP and RIF treatments were included in this review and they were required to meet all of the following six inclusion criteria. These inclusion criteria were: (1) interventions: the intrauterine infusion of PRP around the time of ET; (2) the controls: having undergone no treatment or other treatments; (3) the population were diagnosed as having had an RIF; (4) the pregnancy outcomes were confirmed; (5) only English language studies were accepted; (6) having an endometrial thickness of ≥7 mm. Other studies such as case reports, animal experiments, cell experiments, research that was irrelevant to PRP and RIF, bibliometric analyses, poor-quality literature, self-pro-post studies, reviews and abstracts were excluded. Clinical pregnancy rates were the primary outcome, and the secondary results were the live birth rates (LBR), positive serum β-HCG rates 14 days after the ET, the implantation rates, and the miscarriage rates. Clinical pregnancy was defined as a pregnancy that was diagnosed by ultrasonographic visualization of one or more gestational sacs or definitive clinical signs of pregnancy. In addition to an intra-uterine pregnancy, it includes a clinically documented ectopic pregnancy [23].

### 2.3. PRP Protocols

Peripheral venous blood was drawn using a syringe containing anticoagulant solution and centrifuged immediately to separate the red blood cells. The liquid supernatant was centrifuged again to separate the plasma and obtain the PRP with platelets. In addition, the peripheral blood was divided into three layers after the first centrifugation in the studies of Yangying Xu et al., and Mahvash Zargar et al., and the top layer was centrifuged again to obtain the PRP [3,6].

### 2.4. Statistical Analysis

We extracted the relative reproductive outcomes from the included studies. The RR and corresponding 95% CI for each study endpoint were calculated by the Mantel–Hansel method using the fixed-effects model between the intervention group and the control group according to the Review Manager 5.4. The heterogeneity of the studies was assessed graphically with forest plots and statistically by chi-square-based Q statistic and I^2^ value. Heterogeneity, the statistical measure of homogeneity, was considered significant at a *p*-value of <0.05 in Q-test or I^2^ > 50%. According to the dosage of PRP (0.5–1 mL group and ≥1 mL group) and the study design (RCT and cohort), a subgroup analysis was used to identify the possible sources of heterogeneity for the effect of an intrauterine infusion of the PRP. A funnel plot was used to assess the reporting bias.

## 3. Results

### 3.1. Summary of Literature Research and Description of Studies

In total, 227 publications (38 from PubMed, 57 from Cochrane Library, 65 from Embase, and 67 from Web of Science) were searched using the terms above. All of the citations were imported into Endnote to eliminate 113 duplicates. Next, we scrutinized two case reports, eight animal or cell studies, thirty-one studies that were not part of the literature about PRP and RIF, one bibliometric analyses, twenty-one reviews, thirty-seven abstracts, one poor-quality study, and three self-pro-post control studies, and ten studies were finally analyzed. The flow diagram of the literature search and selection of studies is shown in Figure 1. The details of the selected studies are showed in Appendix A. Apart from the review, abstract, a bibliometric analysis, and the studies that were not related to PRP and RIF, we read the full texts to censor the studies. The exclusion reasons are presented in Appendix A.

### 3.2. Study Characteristics

Appendix A lists the characteristics of all of the selected studies. Eight studies were conducted in Iran [4,5,6,24,25,26,27,28], and another two studies were conducted, respectively, in China and India [3,29]. Six studies were RCTs, and four were cohorts. All of the control groups received no treatment, except one, which was compared with the granulocyte colony stimulating factor (GCSF) [26]. The population of one half of the studies was infused with a PRP of less than 1 mL; the other half received ≥1 mL. Only one study did not give an accurate time of intrauterine perfusion of the PRP; the others were conducted at 2–3 days before the ET [29].

### 3.3. Risk of Bias Assessment

The summary of the risks of the bias assessments are shown in Appendix A. For a random sequence generation, five studies were judged to have low risks of bias, three were judged to have high risks, and two were judged to have unclear risks. Four of the trials were assessed as having high risks of bias, the others had unclear risks for the allocation concealment. Another bias, including the blinding of the participants and personnel, the blinding of outcome assessment, incomplete outcome data, selective reporting, and other bias were judged as low risks. All of the research selected the population from the same community sample and provided exacted diagnosis criteria for the interesting reproductive outcomes.

### 3.4. Clinical Pregnancy Rates

The summary results included 1629 participants (795 cases and 834 controls) from 10 studies. As shown in Figure 2, the comparison of the clinical pregnancy rates indicated that the intrauterine infusion of the PRP had a better effect on the clinical pregnancy outcomes when it was compared with that of the control group according to fixed-effects model analysis (n = 10, risk ratio (RR) = 1.79, 95% confidence intervals (CI): 1.55, 2.06; *p* < 0.01, I^2^ = 40%). In Appendix A, the funnel plot appears to be asymmetric, with some missingness at the lower left portion of the plot suggesting a possible publication bias, which means that some positive results from the small sample of studies with low precision were not published. The sensitivity analysis showed that the estimates of the summary RR ranged from 1.64 (95% CI: 1.39, 1.93) to 1.90 (95% CI: 1.63, 2.21), which meant that the pooled results were not overly influenced by a single study.

Considering that we used *p*-values < 0.1 and there was an I^2^ = 0.45% of heterogeneity testing in all of the cases and controls, a corresponding subgroup analysis was performed to recover what caused heterogeneity of the two groups. The PRP dosage and study design may be the sources of it, and so we analyzed the traits of the selected studies. When the women were treated at a 0.5–1 mL dose of PRP (n = 5, RR = 2.24, 95% CI: 1.80, 2.79; *p* < 0.01, I^2^ = 0%), the effect size was stronger than ≥1 mL (n = 5, RR = 1.48, 95% CI: 1.22, 1.80; *p* < 0.01, I^2^ = 23%), and more patients benefited in terms of the clinical pregnancy rates (Figure 3). Within the subgroups, there is only negligible heterogeneity. In Figure 4, the cohort studies (n = 4, OR = 1.79, 95% CI: 1.32, 2.43; *p* < 0.01, I^2^ = 36%) had a weaker effect when they were compared with the RCTs (n = 6, OR = 2.98, 95% CI: 2.29, 3.88; *p* < 0.01, I^2^ = 14%) in terms of the pregnancy outcomes, and the heterogeneity was not significant within the subgroups. With an asymmetric funnel plot of the risk ratios of the clinical pregnancy outcomes, we evaluated the reporting bias again, excluding the possible confounding factors, study design, and PRP dosage (Appendix A). In the study design subgroup, the funnel plot was symmetric, and the asymmetric reporting bias that is mentioned above could result from that.

### 3.5. Live Birth Rates

In Figure 5, four trials including 878 patients (433 cases and 445 controls) demonstrated that the live birth rates in patients that underwent an intrauterine infusion of PRP significantly increased when they were compared with the controls (n = 4, RR = 2.92, 95% CI: 2.22, 3.85; *p* < 0.01, I^2^ = 83%). We removed one study with high heterogeneity; the live birth rates remained statistically different between the two groups (n = 3, RR = 1.9, 95% CI: 1.39, 2.59; *p* = 0.43, I^2^ = 0) (Appendix A) [4].

### 3.6. Positive Serum β-HCG Rates on 14 Days after ET and Implantation Rates

Six studies with 931 participants (440 cases and 491 controls) compared the serum β-HCG 14 days after the ET, and the positive rates of the experimental group were significantly higher than those of the control group (n = 8, RR = 1.77, 95% CI: 1.54, 2.03; *p* < 0.01, I^2^ = 36%) (Figure 6). One study was excluded owing to us not finding the test time of β-HCG [24]. The implantation rates were also evaluated by three studies involving 1061 women (549 cases and 512 controls), and it was shown that the intervention of PRP significantly increased the rates of the implantation when they were compared to those of the controls (n = 3, RR = 1.74, 95% CI: 1.34, 2.26; *p* < 0.01, I^2^ = 0) (Figure 7). We excluded two studies because the data of one study could not be extracted, and the calculation formula of the other data was controversial [5,28]. Both the higher positive serum β-HCG rates 14 days after the ET and the implantation rates proved that the intrauterine infusion of PRP increased the possibility of implantation in comparison with that resulting from no treatment or other treatments for the women with RIF.

### 3.7. Miscarriage Rates

Next, we assessed the effect of PRP on the miscarriage rates for patients that experienced an RIF in six studies (440 cases and 491 controls). There was no significant difference between the intervention and the control group, and it seemed that no heterogeneity factor disturbed its analysis process (n = 6, RR = 1.04, 95% CI: 0.72, 1.51; *p* = 0.83, I^2^ = 0%) (Figure 8).

## 4. Discussion

In this study, 10 studies were included to assess the efficacy of the intrauterine perfusion of RPR for 1629 patients (795 cases and 834 controls) with unexplained repeated implantation failures, who were undergoing an embryo transfer. The endometrial thickness on the day of the HCG in our included population was no less than seven millimeters, and so the effect of a thin endometrium on the pregnancy outcomes was excluded.

In this meta-analysis, a fixed-effects model was used to assess the effect of the PRP in comparison with the control on the clinical pregnancy rates, live birth rates, implantation rates, and positive serum β-HCG 14 days after the ET and the miscarriage rates. RR and 95% CI showed that PRP group had better outcomes in terms of clinical pregnancy, live birth, implantation, and positive β-HCG 14 days after the embryo transfer. These results were consistent with PRP increasing the chance of pregnancy and delivery for females with RIF [4,25,27,28]. However, the PRP did not show significant advantages in improving the miscarriage rates, which was also proven in the findings of previous epidemiological studies [5,27,28]. Preferable embryo implantation, pregnancy, and live birth results suggest that the intrauterine infusion of PRP facilitates the embryo transfer in patients that have undergone an RIF. Unfortunately, not all of the included studies gave the research outcomes that we needed.

A subgroup analysis was carried out to disclose the reasons for the heterogeneity in the clinical pregnancy results between the two groups. The advantages of PRP were in improving the clinical pregnancy after the subgroup analysis regarding the PRP dosage (0.5–1 mL versus ≥1 mL) and the study design (cohort versus RCT). The effect size of 0.5–1 mL dose of PRP was stronger than a ≥1 mL one was, and more patients benefited in terms of the clinical pregnancy rates when they were treated with 0.5–1 mL dose of PRP. However, we did not think this result meant that a 0.5–1 mL dose of PRP was more suitable for patients with RIF because the required components were not reported in the included studies.

Autologous platelet-rich plasma is a platelet-rich whole blood extract without red or white blood cells. Because it is extracted from the patient’s own peripheral blood, it is easy to obtain, inexpensive, and reduces the occurrence of an immune rejection. PRP is rich in growth factors, typically the platelet-derived growth factor (PDGF), the transforming growth factor (TGF), the vascular endothelial growth factor (VEGF), and the epidermal growth factor [30,31]. It induces regeneration and differentiation, accelerates endometrial damage repairment, and has anti-inflammatory properities [31,32,33,34,35]. Siwen Zhang et al., described in his study that PRP had a strong effect on endometrial regeneration, uterine damage restoration, and the increased proliferation of stromal cells, progenitor cells, and the vessel density of the endometrium [35]. These effects may explain why PRP improves the pregnancy outcomes in IVF/ICSI patients with RIF.

The heterogeneity analysis of the clinical pregnancy rates seemed high, although there were no significant differences. We designed a subgroup analysis regarding the PRP dosage, and the study design and reduced heterogeneity indicated that these two were confounding factors. The asymmetrical funnel plot indicated that there was a possible reporting bias. However, it changed to a symmetrical plot in the subgroup analysis of the study design, which suggested that the reason for the asymmetry may be attributed to the types of trials.

As there is no unified method to extract PRP from the peripheral blood, we have not yet obtained detailed key information such as the concentrations and activity of the platelets and growth factors to determine how PRP works. Thus, we cannot provide a reasonable explanation for the relationship between the PRP dosage and the IVF/ICSI outcomes in terms of RIF.

The design of the study and the outcomes that were measured affect the strength of the evidence according to evidence-based medicine (EBM) [36]. Randomized controlled trials are usually considered to be more convincing than cohort ones owing to the former’s objectivity. Involving studies that included six RCTS and only four cohort studies also made this meta-analysis more objective and convincing.

It is the first that a meta-analysis and review was conducted to assess the intrauterine infusion of PRP to the influence reproductive outcomes of unexplained RIF patients undergoing an embryo transfer. In this review, we excluded the influence of endometrium thickness and childbearing age because some studies found that an endometrial thickness of ≤7 mm and a high reproductive age (all of the women included were below 40 years) may negatively impact the pregnancy results [37,38,39,40]. In addition, apart from one study that did not mention the exact time of the PRP infusion, all of the other populations used the PRP protocol two or three days before the ET [2]. This meta-analysis may provide a referable time point to initiate PRP.

It is undeniable that we did not give a strict definition of repeated implantation failures owing to a lack of consensus, and we included all of the patients who were diagnosed with a RIF by referring to the local criteria. Except for this, heterogeneity and a reporting bias existed in the process of the analysis, which had an impact on the credibility and objectivity of the article, but only based on the existing data, we may not completely rule out the occurrence of heterogeneity and bias. Meanwhile, the risk of bias graph and risk of bias summary also showed some risks in the selection bias. In addition, two studies involved some participants undergoing a fresh embryo transfer, however, we cannot extract them from the statistics to conduct a subgroup analysis. We cannot draw a conclusion as to which embryo transfer protocol benefits more from PRP. At the same time, our included studies did not report whether euploid tests were carried out or not. We suggest that the following studies can list more detailed data if they include more than one embryo transfer method.

Most suspiciously, most of the articles that we included did not control the variables. The other controls did not take any treatment besides one, which involved infusing GCSF [26]. However, during the intrauterine infusion PRP the insertion and removal of tubes is needed to finish the treatments. If the controls only took measures of no treatment, we were unable to assess the effect of the mechanical manipulation of the uterine cavity on the pregnancy outcomes [41,42].

A very limited number of studies were considered for an overly large number of confounding factors in this meta-analysis, and a further RCT should be conducted to prove these results.

## 5. Conclusions

This systematic review and meta-analysis proved that the intrauterine infusion of PRP has a positive effect on the pregnancy results for patients with unexplained repeated implantation failures, who are undergoing an embryo transfer.

## Figures and Tables

**Figure 1 jcm-11-06753-f001:**
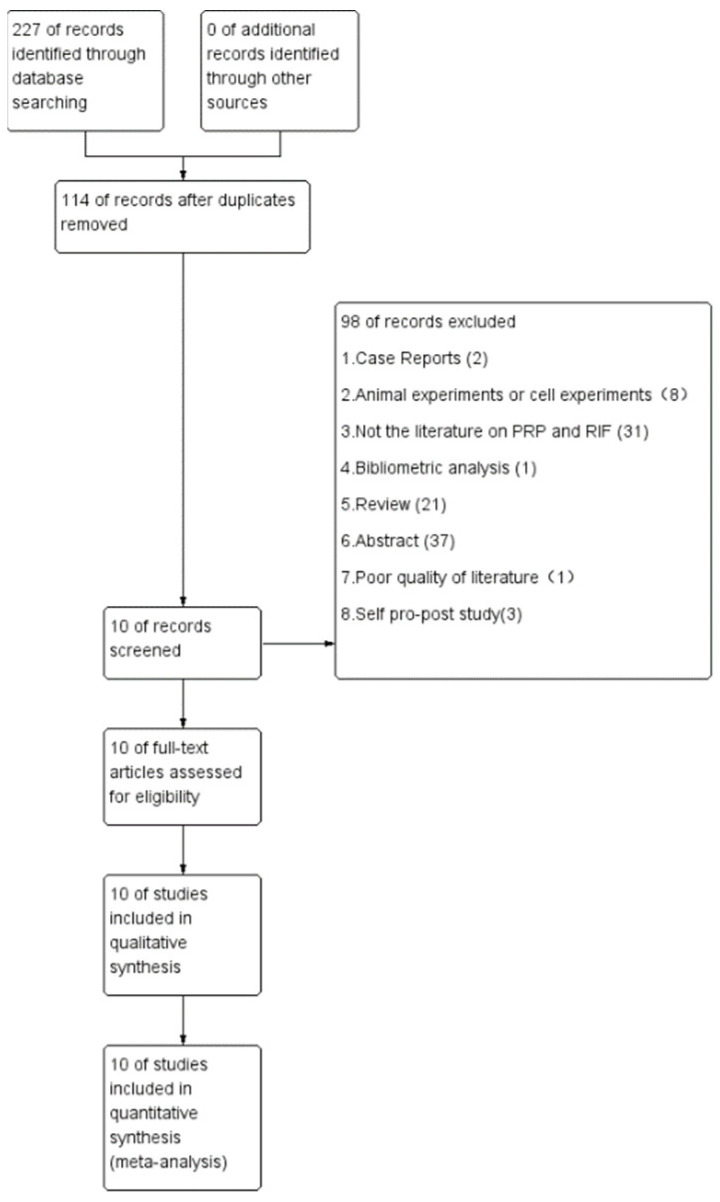
Flow diagram of the selection process.

**Figure 2 jcm-11-06753-f002:**
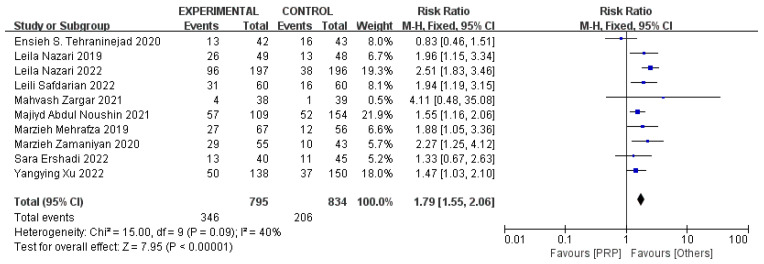
Forest plot of RR, 95% CI, and heterogeneity in studies that evaluated the risk of clinical pregnancy in interventions versus controls [3,4,5,6,24,25,26,27,28,29].

**Figure 3 jcm-11-06753-f003:**
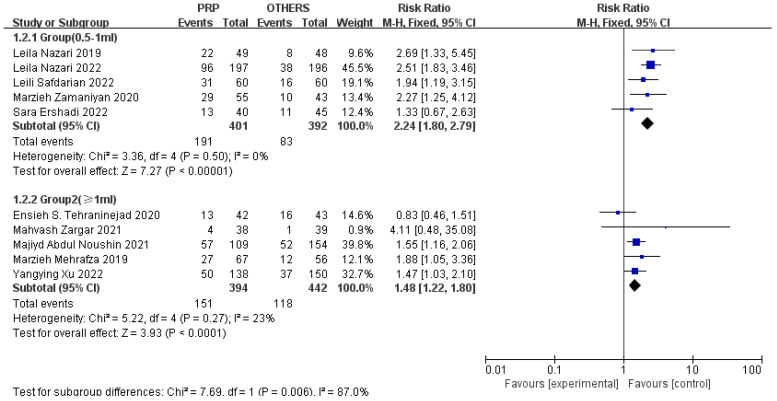
Forest plot of RR, 95% CI, and heterogeneity in studies that evaluated the risk of clinical pregnancy in interventions versus controls regarding doses of PRP [3,4,5,6,24,25,26,27,28,29].

**Figure 4 jcm-11-06753-f004:**
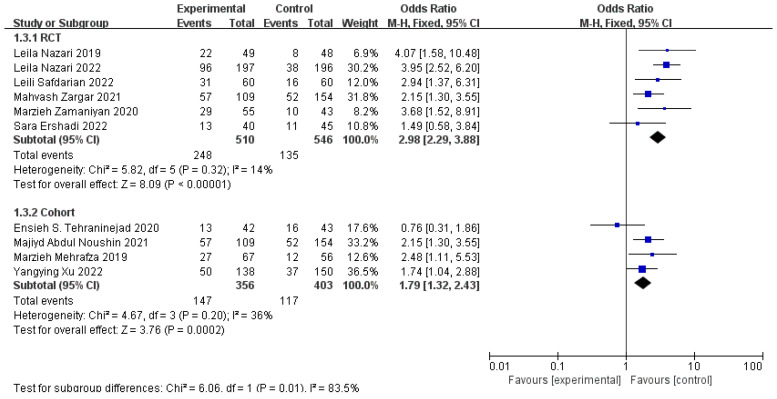
Forest plot of OR, 95% CI, and heterogeneity in studies that evaluated the risk of clinical pregnancy in interventions versus controls regarding study design [3,4,5,6,24,25,26,27,28,29].

**Figure 5 jcm-11-06753-f005:**
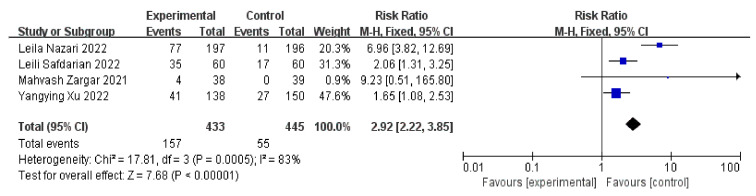
Forest plot of RR, 95% CI, and heterogeneity in studies that evaluated the risk of live birth rates in interventions versus controls [3,4,6,27].

**Figure 6 jcm-11-06753-f006:**
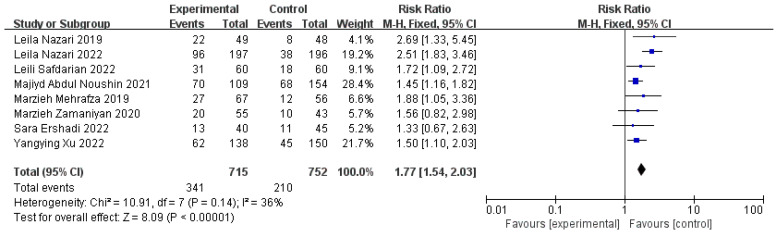
Forest plot of RR, 95% CI, and heterogeneity in studies that evaluated the risk of positive serum β-HCG rates 14 days after ET in interventions versus controls [3,4,5,25,26,27,28,29].

**Figure 7 jcm-11-06753-f007:**
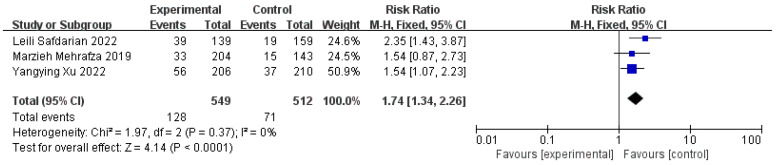
Forest plot of RR, 95% CI, and heterogeneity in studies that evaluated the risk of implantation rates after ET in interventions versus controls [3,26,27].

**Figure 8 jcm-11-06753-f008:**
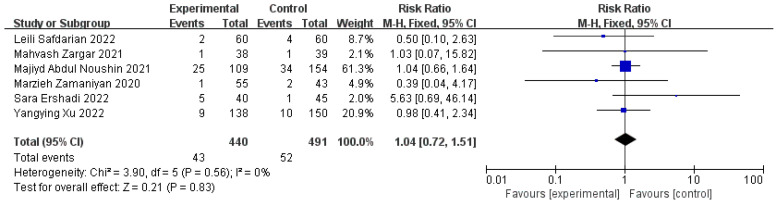
Forest plot of RR, 95% CI, and heterogeneity in studies that evaluated the risk of miscarriage rates after ET in interventions versus controls [3,5,6,27,28,29].

## Data Availability

Not applicable.

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
