# Peer review of "Efficacy of Autologous Intrauterine Infusion of Platelet-Rich Plasma in Patients with Unexplained Repeated Implantation Failures in Embryo Transfer: A Systematic Review and Meta-Analysis"

_jcm, 2022, doi:10.3390/jcm11226753_

Round 1

Reviewer 1 Report

Reviewer’s report

Title:

Date: 4 October 2022

Manuscript ID: jcm-1931155

In the manuscript titled “Efficacy of Autologous Intrauterine Infusion of Platelet-Rich Plasma in Patients with Unexplained Repeated Implantation Failures on Embryo Transfer: A Systematic Review and Meta-analysis,” the authors evaluated the role of PRP in women with repeated implantation failure undergoing embryo transfer. The manuscript may be useful for advancement of the field; however, it  requires significant changes . Most notably, the paper needs significant grammatical edits and inclusion of the data table described in the results section. Additional detailed comments are as follows:

·         Abstract:

o   Controversial outcomes is unclear in the abstract. As it is the first sentence of the abstract, more context is needed.

·         Introduction:

o   Much of the introduction needs to be rewritten as the content may be well organized and provide useful context for the meta-analysis, but it is very difficult to understand due to the quality of the English.

o   The introduction of PRP in the introduction section should include more citations and a more robust discussion of how PRP came to be used in female infertility cases as well as why it may be beneficial for improving implantation rates/embryo transfer.

·         Methods:

o   The literature search appears adequate.

o   The inclusion and exclusion criteria are confusing. There are 6 listed “criteria” which I believe to be inclusion criteria but it is unclear if all 6 needed to be met or if any combination of these led to inclusion in the paper. There is also a sentence that reads “The randomized controlled trials and cohorts between PRP and RIF treatment were included in this review” but it is unclear if all RCTs/cohort studies were included, if only RCTs/cohort studies were included, or if only RCTs/cohort studies that met the following 6 criteria were included.

·         Results:

o   Table 1 is mentioned but not provided in the document

o   No studies in the United States or Europe were included, was this on purpose? If so, please describe why.

·         Discussion:

o   The first few paragraphs of the discussion section are the strongest of the paper.

o   The synthesis of ideas in the discussion section is and arrival at a conclusion is unclear. An additional paragraph describing how they reached the conclusion is necessary.

·         An English grammar check is highly recommended & the manuscript needs to be rewritten by an English scholar with a focus on use of colloquial language and verb/tense agreement.

o   For example, the second sentence of the paper describes “female fertility” but I think the authors meant infertility and the rest of the sentence does not make sense

Author Response

Response to Reviewer #1 Comments

Comments: In the manuscript titled “Efficacy of Autologous Intrauterine Infusion of Platelet-Rich Plasma in Patients with Unexplained Repeated Implantation Failures on Embryo Transfer: A Systematic Review and Meta-analysis,” the authors evaluated the role of PRP in women with repeated implantation failure undergoing embryo transfer. The manuscript may be useful for the advancement of the field; however, it requires significant changes. Most notably, the paper needs significant grammatical edits and the inclusion of the data table described in the results section. 

Response: Thank you for putting time and effort into reviewing the previous version of the manuscript. The suggestions have enabled us to improve our work.

Abstract:

Q1:    Controversial outcomes is unclear in the abstract. As it is the first sentence of the abstract, more context is needed.

Response 1: Special thanks to you for your good comments. I am sorry for not providing the details of the controversial outcomes owing to the word limit of the abstract. In the literature reviewed, different research teams reached different conclusions on the impact of PRP on assisted reproductive outcomes in patients with repeated implantation failure. For example, Yangying Xu et al. found intrauterine infusion of PRP could significantly improve clinical pregnancy rates and live birth rates for patients with RIF and undergoing embryo transfer, however, Mahvash Zargar et al. did not think they would benefit from it. So, we pointed the controversial outcomes in the first sentence of the abstract and the related studies are as follows:

Xu Y, Hao C, Fang J, Liu X, Xue P, Miao R. Intrauterine Perfusion of Autologous Platelet-Rich Plasma Before Frozen-Thawed Embryo Transfer Improves the Clinical Pregnancy Rate of Women with Recurrent Implantation Failure. Front Med (Lausanne). 2022 Mar 29;9:850002. doi: 10.3389/fmed.2022.850002. PMID: 35425782; PMCID: PMC9001903.

Zargar M, Pazhouhanfar R, Najafian M, Choghakabodi PM. Effects of intrauterine autologous platelet-rich plasma infusions on outcomes in women with repetitive in vitro fertilization failures: a prospective randomized study. Journal: Review. Clinical and experimental obstetrics & gynecology. 2021;48(1):180‐185. doi:10.31083/j.ceog.2021.01.2131.

Instruction:
Q2: Much of the introduction needs to be rewritten as the content may be well organized and provide useful context for the meta-analysis, but it is very difficult to understand due to the quality of the English.

Response 2: Thank the Reviewer for this precious comment. Special thanks to you for your good comments. We regret there were problems with the English. The paper has been carefully revised by a native English speaker of MDPI to improve the grammar and readability. And the rewritten introduction is as follows:

  A considerable proportion of couples worldwide suffer from infertility. The common reasons for female infertility issues include ovulation disorders, fallopian-related disorders, uterine disorders, and unexplained infertility. Although assisted reproductive technology (ART) has rapidly developed in recent years, the causes and treatments of repeated implantation failures (RIF, recurrent implantation failures) continue to plague reproductive specialists. There is no accepted formal definition for RIF due to the fact that RIF was initially considered a rather heterogeneous entity. Some studies defined it as a failure of the concept after three or more embryo transfers (ETs) with high-quality embryos. It was, however, also accepted that RIF was considered as a disorder that infertility women with at least two ET failures. Even so, quite a few specialists suggested a more complete working definition taking into account maternal age, the number of embryos transferred, and the number of cycles completed. Accumulative data have clarified that most of the etiology does not have the evidence base for a generalized application to be suggested by the relevant societies. The etiology of RIF was currently attributed to dysfunction of the embryo and the endometrium, and Antonis Makrigiannakis et al. also divided it into several factors, namely anatomy, immunology, dysbiotic microbiota, and unexplained reasons, etc., in the review. Moreover, this team also described some of the main treatment protocols including endometrium injury, human chorionic gonadotropin, peripheral blood mononuclear cells, and platelet-rich plasma (PRP). RIF is a constant challenge in ART with a burden on health providers and infertile couples.

  The extracted protocols of platelet rich-plasm also remain inconsistent with no consensus around the world. It is generally defined as an autologous blood-derived concentrate of platelets from peripheral blood that has a platelet count 35 times higher than the baseline concentration with growth factors and other cytokines such as transforming growth factor beta (TGF-β) and interleukin-1β (IL-1β). Presently, PRP is widely used in knee osteoarthritis, erectile dysfunction, medical dermatology, periodontal regeneration and facial rejuvenation.

  In reproductive medicine, poor ovarian reserve, premature ovarian failure, and thin endometrium have been the main areas of research on PRP by intraovarian injection or intrauterine infusion. Studies found that PRP had high growth factor and cytokine concentrations, which were considered very important in cell proliferation, chemotaxis, cell differentiation, regeneration, and angiogenesis. Fady I Sharara et al. reviewed the previous literature on the effects of autologous PRP in reproductive medicine, finding that PRP can increase the endometrial thickness in thin endometrium. These reasons may explain why PRP could improve the implantation outcomes and be beneficial for embryo transfer.

  There has been a surge in high-level studies investigating PRP for implantation failures. Therefore, in this systematic review and meta-analysis, we aimed at investigating the effect of intrauterine infusion of autologous PRP in women with unexplained RIF undergoing IVF/ICSI cycles.

Q3: The introduction of PRP in the introduction section should include more citations and a more robust discussion of how PRP came to be used in female infertility cases as well as why it may be beneficial for improving implantation rates/embryo transfer.

Response 3: Special thanks to you for your good comments. We looked up some more references listed as citations that added the role of PRP in female infertility. We found PRP was mainly used in poor ovarian reserve, premature ovarian failure, and thin endometrium in recent years [1-3]. We also added how did PRP function in improving implantation rates/embryo transfer in the introduction section apart from the part originally elaborated in the discussion section. Studies found that PRP had high growth factor and cytokine concentrations, which were considered very important in cell proliferation, chemotaxis, cell differentiation, regeneration, and angiogenesis [4-5]. Fady I Sharara et al. reviewed the previous literature on the effects of autologous PRP in reproductive medicine, finding that PRP can increase the endometrial thickness in thin endometrium [6]. These reasons may explain why PRP could improve the implantation outcomes and be beneficial for embryo transfer. (Page 2, Line 68-76 in the highlighted version)

The added references are as follows:

  1. Chang Y, Li J, Wei L-N, Pang J, Chen J, Liang X. Autologous platelet-rich plasma infusion improves clinical pregnancy rate in frozen embryo transfer cycles for women with thin endometrium. Medicine (Baltimore). 2019;98(3):e14062. doi:10.1097/MD.0000000000014062
  2. Cakiroglu Y, Saltik A, Yuceturk A, et al. Effects of intraovarian injection of autologous platelet rich plasma on ovarian reserve and IVF outcome parameters in women with primary ovarian insufficiency. Aging (Albany NY). 2020;12(11):10211-10222. doi:10.18632/aging.103403
  3. Hsu C-C, Hsu L, Hsu I, Chiu Y-J, Dorjee S. Live Birth in Woman With Premature Ovarian Insufficiency Receiving Ovarian Administration of Platelet-Rich Plasma (PRP) in Combination With Gonadotropin: A Case Report. Front Endocrinol (Lausanne). 2020;11:50. doi:10.3389/fendo.2020.00050
  4. Amable PR, Carias RBV, Teixeira MVT, et al. Platelet-rich plasma preparation for regenerative medicine: optimization and quantification of cytokines and growth factors. Stem Cell Res Ther. 2013;4(3):67. doi:10.1186/scrt218
  5. Sánchez-González DJ, Méndez-Bolaina E, Trejo-Bahena NI. Platelet-rich plasma peptides: key for regeneration. Int J Pept. 2012;2012:532519. doi:10.1155/2012/532519
  6. Sharara FI, Lelea LL, Rahman S, Klebanoff JS, Moawad GN. A narrative review of platelet-rich plasma (PRP) in reproductive medicine. Review. Journal of Assisted Reproduction and Genetics. 2021;38(5):1003-1012. doi:10.1007/s10815-021-02146-9

Methods:

Q4:  The literature search appears adequate.

Response 4: Thanks for your comment.

Q5: The inclusion and exclusion criteria are confusing. There are 6 listed “criteria” which I believe to be inclusion criteria, but it is unclear if all 6 needed to be met or if any combination of these led to inclusion in the paper. There is also a sentence that reads “The randomized controlled trials and cohorts between PRP and RIF treatment were included in this review” but it is unclear if all RCTs/cohort studies were included, if only RCTs/cohort studies were included, or if only RCTs/cohort studies that met the following 6 criteria were included.

Response 5: Thank you very much for your advice. I apologize for not clarifying the inclusion and exclusion criteria. Included studies were required to meet 6 inclusion criteria. In this sentence, “The randomized controlled trials and cohorts between PRP and RIF treatment were included in this review”, only RCTs/cohort studies that met the following 6 criteria were included. and we have modified this section as follows: The randomized controlled trials and cohort between PRP and RIF treatment were included in this review, and they were required to meet all of the following six inclusion criteria. These inclusion criteria were 1) interventions: intrauterine infusion of PRP around the time of ET; 2) the controls: no treatment or other treatments; 3) the population were diagnosed as RIF; 4) pregnancy outcomes were confirmed; 5) only English language studies were accepted; 6) endometrial thickness of ≥7 mm.

Results:

Q6:  Table 1 is mentioned but not provided in the document.

Response 6: Thanks for your comment. We will re-upload Table 1. In addition, due to the large number of columns in the table, we changed it to Table S1.(Please see the attachment)

Q7:  No studies in the United States or Europe were included, was this on purpose? If so, please describe why.

Response 7: Thanks for your comment. We did not deliberately exclude European or American studies. We identified potential studies by searching Medline (PubMed), Embase, Cochrane Library, and Web of Science (WOS) with terms in the manuscript, and then selected them by inclusion and exclusion criteria. Finally, the studies that met the criteria did not include European or American ones.

Discussion:

Q8:  The first few paragraphs of the discussion section are the strongest of the paper.

Response 8: Thanks for your comment.

Q9:  The synthesis of ideas in the discussion section is and arrival at a conclusion is unclear. An additional paragraph describing how they reached the conclusion is necessary.

Response 9: Thanks for your comment. In this meta-analysis, a fixed-effects model was used to assess the effect of PRP compared with control on the clinical pregnancy rates, live birth rates, implantation rates, positive serum β-HCG 14 days after ET and miscarriage rates. RR and 95%CI showed that the PRP group had better outcomes in clinical pregnancy, live birth, implantation, and positive β-HCG 14 days after embryo transfer. In addition, we also performed subgroup analysis regarding study design and PRP dosage and the advantages that remained in improving clinical pregnancy rates. 

And the related description is as follows:

In this meta-analysis, a fixed-effects model was used to assess the effect of PRP compared with control on the clinical pregnancy rates, live birth rates, implantation rates, positive serum β-HCG 14 days after ET, and miscarriage rates. RR and 95%CI showed that the PRP group had better outcomes in clinical pregnancy, live birth, implantation, and positive β-HCG 14 days after embryo transfer. These results were consistent with PRP increasing the chance of pregnancy and delivery for females with RIF. However, PRP did not show significant advantages in improving miscarriage rates, which was also proved in the findings of previous epidemiological studies. Preferable embryo implantation, pregnancy, and live birth results suggest that intrauterine infusion of PRP facilitates embryo transfer in patients with RIF. Unfortunately, not all the included studies gave the research outcomes we needed.

Subgroup analysis was carried out to disclose heterogeneity reasons for the clinical pregnancy between the two groups. The advantages of PRP remained in improving clinical pregnancy after subgroup analysis regarding PRP dosage (0.5–1 ml versus ≥1 ml), and the study design(cohort versus RCT). The effect size of 0.5–1 ml dose of PRP was stronger than ≥1 ml and more patients benefited in clinical pregnancy rates when treated with 0.5–1 ml dose of PRP. However, we did not think this result meant 0.5–1 ml dose of PRP was more suitable for patients with RIF because the required components were not reported in the included studies.

Q10:  An English grammar check is highly recommended & the manuscript needs to be rewritten by an English scholar with a focus on use of colloquial language and verb/tense agreement.   For example, the second sentence of the paper describes “female fertility” but I think the authors meant infertility and the rest of the sentence does not make sense.

Response 10: Special thanks to you for your good comments. We regret there were problems with the English. The paper has been carefully revised by a native English speaker of MDPI to improve the grammar and readability.

Reviewer 2 Report

The work is a meta-analysis. It was carried out correctly. Of course, what the authors write about, there is a large possibility of bias , if only because the definition of RIF is difficult. The meta-analysis shows that the infusion of platelet-rich plasma improves the implantation y of embryos, but both the mechanism of action and the real effects require further research.

Author Response

Response to Reviewer #2 Comments

Comments: The work is a meta-analysis. It was carried out correctly. Of course, what the authors write about, there is a large possibility of bias, if only because the definition of RIF is difficult. The meta-analysis shows that the infusion of platelet-rich plasma improves the implantation y of embryos, but both the mechanism of action and the real effects require further research.

Response: Thank you very much for your positive comments. We will carry out further research to explore the mechanism and the real effects.

Reviewer 3 Report

The current systematic review and meta-analysis aimed to evaluate the effect of autologous platelet-rich plasma (PRP) in women with repeated implantation failures (RIF) undergoing embryo transfer (ET). The authors showed a beneficial effect of PRP on clinical pregnancy, live birth and  implantation rates  but not on miscarriage rates. The manuscript is clear . However I think that it is affected by many biases that should further take into consideration and discuss before drawing final conclusions.

Major comments:

1. The definition of RIF is the key critical factor affecting the analysis. The authors shoud clearly state for each consideredstudy  which was the corresponding definition of RIF.

2. Please clarify for each study, maybe in table 1, if ET were fresh or frozen, cleavage stage or blastocyst stage, euploid or untested. Consider if a sub-analysis can be performed.

3. Please discuss pivotal differences in PRP obtaining protocols and/or composition among studies

4. Please add in M&M the definition for the considered outcomes (clinical pregnacy, miscarriage and live birth rates)

5. The authors shold be discuss results with more caution. The meta-analysis considered a very limited number of studies for a too large number of confounding factors. Further RCT should be conducted before drawing final conclusions.

Minor comments:

1. Manuscript layout should deeply revised to make it more readable before consideration for publication

Author Response

Q1: The definition of RIF is the key critical factor affecting the analysis. The authors shoud clearly state for each considered study which was the corresponding definition of RIF.

Response 1: Special thanks to you for your good comments. We couldn't agree with you more. Unfortunately, there is no accepted formal definition for RIF due to the fact that RIF was initially considered a rather heterogeneous entity. Some studies defined it as a failure of the concept after three or more embryo transfers with high-quality embryos. It was, however, accepted that RIF was considered as a disorder that infertility women with at least two ET failures. Therefore, there was no single sentence specifically defining RIF in our manuscript, and the characteristics of RIF in all of the included studies were shown by the Population of Table S1.  (Please see the attachment)

Q2: Please clarify for each study, maybe in table 1, if ET were fresh or frozen, cleavage stage or blastocyst stage, euploid or untested. Consider if a sub-analysis can be performed.

Response 2: Thanks for your comment. We have added Transfer type, time of embryo transfer, if genetic testing of the included population were done in Table S1(Transfer type, Time of embryo transfer). As for the euploid or untested, there was no related record in the included studies. We have to admit this is the limitation of our included studies. I am sorry for not subgroup analysis about these classifications because Mahvash Zargar 2021, Sara Ershadi 2022, and Marzieh Mehrafza 2019 included both FET and fresh embryo transfer and Yangying Xu 2022, Majiyd Abdul Noushin 2021 and Marzieh Mehrafza 2019 implanted embryos of both cleavage stage or blastocyst stage.

Q3:  Please discuss pivotal differences in PRP obtaining protocols and/or composition among studies.

Response 3: Thank you for the comment. PRP was prepared from autologous blood using a two-step process. Peripheral venous blood was drawn in the syringe containing Anticoagulant solution and centrifuged immediately to separate the red blood cells. The liquid supernatant was centrifuged again to separate plasma and obtain the PRP with platelets at about 4–5 times higher in concentration than circulating blood. In addition, peripheral blood was divided into three layers after the first centrifugation in the studies of Yangying Xu et al. and Mahvash Zargar et al. and the top layer remained to centrifuge again. In all the included studies of this meta-analysis, and what the researchers collected was the top layer after centrifugation. The differences of PRP obtaining protocols mainly showed in primary volume of drawing peripheral venous blood, volume of Anticoagulant solution, the first and second centrifugal speed and time as shown in the following table. It is reasonable to assume that the components of PRP in the included literature are similar. We also showed the compositions of PRP mentioned in the included studies in the table. The difference of the PRP compositions was mainly the concentrations of platelet in PRP. Otherwise, two studies also mentioned the concentrations of lymphocyte of PRP, however, it is not easy to tell the difference of it because the other eight studies did not record these componients. 

Volume of peripheral venous blood (mL)

Volume of Anticoagulant solution (mL)

First centrifugal speed and time

Second centrifugal speed and time

Platelet concentration

Yangying Xu           2022

20

NOT MENTIONED

500 × g for 10 min

500 × g for 10 min

1513.45 ± 322.18 × 109 /L

Leili Safdarian          2022

8.5

1.5

1600 rpm for 10 minutes

3500 rpm for 6 minutes

4–5 times more than peripheral blood (2000 lymphocyte/µL)

Leila Nazari            2022

8.5

1.5

1200 rpm for 12 min

3300 rpm for 5 min

4–5 times more than peripheral blood

Mahvash Zargar         2021

8.5

1.5

12,000X g for 10 min

12,000X g for 10 min

NOT MENTIONED

Marzieh Zamaniyan      2020

17.5

2.5

1200 rpm for 12 min

3300 rpm for 7 min

4–7 times more than peripheral blood

Ensieh S. Tehraninejad   2020

10

1.5

1,200 rpm for 10 min

3300 rpm for 5 min

4–5 times more than peripheral blood

Majiyd Abdul Noushin   2021

10

NOT MENTIONED

1,200 rpm for 10 min

3,000 rpm for 10min

NOT MENTIONED

Leila Nazari            2019

8.5

1.5

1,200 rpm for 10 min

3300 rpm for 5 min

4–5 times more than peripheral blood

Sara Ershadi           2022

8

2.5

1200 rpm for 12 min

3300 rpm for 7 min

4–5 times more than peripheral blood

Marzieh Mehrafza      2019

8.5

1.5

NOT MENTIONED

NOT MENTIONED

4–5 times more than peripheral blood (2000 lymphocyte/µL)

Q4: Please add in M&M the definition for the considered outcomes (clinical pregnacy, miscarriage and live birth rates)

Response: Thank you for the comment. We couldn't agree with you more.  Clinical pregnancy is defined as the presence of fetal heartbeat or the gestational sac in transvaginal ultrasonography 4-6 weeks after embryo transfer. Live birth is defined as the delivery of a live-born child after 24 weeks of gestational age. Miscarriage is defined as s a fetal loss before 20 weeks of gestation. We have added in Methods the definition for the considered outcomes.

Q5The authors should be discuss results with more caution. The meta-analysis considered a very limited number of studies for a too large number of confounding factors. Further RCT should be conducted before drawing final conclusions.

Response: Thank you for the comment. This is really the limitation of our study and RCT is worthy of consideration further to prove these results.

Round 2

Reviewer 1 Report

The authors answered all the comments properly.

Reviewer 3 Report

The authors are addressed my comments accordingly. The manuscript can be considered fo further publication.